# SARS-CoV-2 hijacks folate and one-carbon metabolism for viral replication

Yuchen Zhang[1,2,3,4,8], Rui Guo[1,2,3,8], Sharon H. Kim[1,5,8], Hardik Shah [1,5,8], Shuting Zhang[1], Jin Hua Liang[1,2,3], Ying Fang[6], Matteo Gentili [1], Colin N. O' Leary[7], Steven J. Elledge[7], Deborah T. Hung [1], Vamsi K. Mootha[1,5,9✉] & Benjamin E. Gewurz [1,2,3,9✉]

The recently identified Severe Acute Respiratory Syndrome Coronavirus 2 (SARS-CoV-2) is the cause of the COVID-19 pandemic. How this novel beta-coronavirus virus, and coronaviruses more generally, alter cellular metabolism to support massive production of ~30 kB viral genomes and subgenomic viral RNAs remains largely unknown. To gain insights, transcriptional and metabolomic analyses are performed 8 hours after SARS-CoV-2 infection, an early timepoint where the viral lifecycle is completed but prior to overt effects on host cell growth or survival. Here, we show that SARS-CoV-2 remodels host folate and one-carbon metabolism at the post-transcriptional level to support de novo purine synthesis, bypassing viral shutoff of host translation. Intracellular glucose and folate are depleted in SARS-CoV-2-infected cells, and viral replication is exquisitely sensitive to inhibitors of folate and one-carbon metabolism, notably methotrexate. Host metabolism targeted therapy could add to the armamentarium against future coronavirus outbreaks.

[1] Broad Institute of MIT and Harvard, Cambridge, MA, USA. [2] Division of Infectious Disease, Department of Medicine, Brigham and Women's Hospital, Boston, MA, USA. [3] Department of Microbiology, Harvard Medical School, Boston, MA, USA. [4] Sun Yat-sen University Cancer Center, State Key Laboratory of Oncology in South China, Guangzhou, China. [5] Howard Hughes Medical Institute and Department of Molecular Biology, Massachusetts General Hospital, Boston, MA, USA. [6] Department of Pathobiology, College of Veterinary Medicine, University of Illinois at Urbana-Champaign, Urbana, IL, USA. [7] Division of Genetics, Brigham and Women's Hospital, Department of Genetics, Howard Hughes Medical Institute, Program in Virology, Harvard Medical School, Boston, MA, USA. [8] These authors contributed equally: Yuchen Zhang, Rui Guo, Sharon H. Kim, Hardik Shah. [9] These authors jointly supervised this work: Vamsi K. Mootha, Benjamin E. Gewurz. ✉email: vamsi_mootha@hms.harvard.edu; bgewurz@bwh.harvard.edu

The recently identified severe acute respiratory syndrome coronavirus-2 (SARS-CoV-2) is an enveloped, single-stranded positive-sense RNA coronavirus responsible for COVID-19. SARS-CoV-2 has rapidly spread and has caused nearly a million deaths worldwide in <1 year[1]. Therefore, there is a great need for the identification of novel antiviral targets and therapeutic agents. Yet, much remains to be learned about the lifecycle of SARS-CoV-2, which only recently emerged in 2019[2].

Upon host cell infection, the SARS-CoV-2 replication/transcription complex synthesizes ~30 kilobase viral genomes and highly abundant subgenomic RNAs that serve as templates for viral structural proteins. Subgenomic RNAs are co-terminal, nested mRNAs encoded by a common 5′ leader sequence that is spliced to regions from ~10 kilobases at the 3′ end of the genome. Viral genomes are first produced as negative-strand RNA, which serve as templates for the production of the positive strand virion genomes. Coronavirus positive strand genomes exist at 50- to 100-fold excess of their minus strand counterparts[3] and are coated by the viral nucleocapsid protein (Np) prior to virion assembly[4].

Between 1 and 5 h post-infection by the model betacoronavirus murine hepatitis virus, the percentage of virus-encoded to total cellular protein translation increases by as much as 20,000-fold. Over this time period, the fraction of viral to cellular RNA reaches 90%, much of which is subgenomic RNA[5], underscoring a major transformation in the metabolism of newly infected cells. Recent profiling studies indicate that high levels of SARS-CoV-2 transcripts likewise enable host cell translational machineries to be dominated by production of viral proteins[6], and SARS-CoV-2 also blocks translation of host but not viral mRNAs[7,8]. SARS-CoV-2 encoded proteins assemble a network of double membrane vesicles, leading to the production and secretion of abundant infectious virion[9,10]. Yet, host metabolic pathways that are rapidly subverted by SARS-CoV-2 to support this biosynthesis remain largely unknown.

## Results and discussion

To gain insights into SARS-CoV-2 remodeling of key host metabolic pathways, Vero E6 TMPRSS2 + cells were mock infected, or infected at a multiplicity of infection (MOI) of 2. Vero E6 were used to enable high percentage target cell infection, at an early timepoint (8 h post infection, hpi) prior to the onset of viral cytostatic or cytopathic effects that confound measurements at later timepoints. To control for unwanted effects of lactate and other spent media metabolites in the virus stock, parallel Vero cultures were either infected by SARS-CoV-2 that had been concentrated by membrane filtration, or mock-infected with an equal volume of the virus-depleted flow-through (Fig. 1a). Robust production of viral genomic RNA (gRNA) and nucleocapsid protein (Np) was evident by 8 h post-infection in most cells (Fig. 1b and Supplementary Fig. 1a), consistent with prior estimates of the "eclipse period" time from SARS-CoV adsorption to release of infectious progeny[11].

RNA-seq analysis at 8 hpi revealed robust induction of antiviral genes (e.g. *IFIT1, ZC3HAV1*), NF-κB targets (e.g., *CCL5, CXCL10*), and ER stress response (e.g. *DDIT3, PPP1R15A, GADD45B*), consistent with published analyses[12]. Yet, surprisingly few changes were observed in the abundances of mRNAs encoding metabolic enzymes (Fig. 1c and Table S1), despite a global decrease in host mRNAs in SARS-CoV-2-infected cells (Fig. 1d). Re-analysis of SARS-CoV-1-infected Vero E6 cell mRNA abundance[13] showed strikingly similar results, with minimal remodeling of metabolism pathways evident at the mRNA level at 8 hpi, suggesting that this property may be conserved across SARS coronaviruses (Supplementary Fig. 1b).

Curiously, nearly all of the mitochondrial DNA (mtDNA) encoded transcripts related to oxidative phosphorylation (OXPHOS) were elevated, while nuclear genome encoded OXPHOS transcripts were all modestly decreased (Supplementary Fig. 1c), a gene expression pattern that has been previously been associated with ATP depletion (Supplementary Data 1)[14].

To cross-compare SARS-CoV-2-mediated transcriptional and metabolic changes within the same cells, metabolomic analyses of spent media and cell pellets were performed in parallel using six replicates (Fig. 1a and Supplementary Fig. 2a). Quantitative production and consumption analyses of spent media metabolites found subtle, mostly non-significant changes between infected and mock-infected cells at this early timepoint (Supplementary Fig. 2b and Supplementary Data 2). Only consumption of pyruvate and release of aspartate scored as significantly increased in SARS-CoV-2-infected cells, albeit mildly with a fold-change difference of 0.06 and 0.16 respectively.

By contrast, pronounced differences in specific intracellular metabolites were already observed at this early timepoint (Fig. 1e, f). The sum of all identified metabolite peaks (a proxy for the total ion current) was unchanged, suggestive of equivalent loading of samples for LC-MS analysis[15] (Supplementary Fig. 2c and Supplementary Data 3). While most amino acids were depleted by SARS-CoV-2 infection, aspartate and asparagine were the most upregulated (Fig. 1f). Amino acids may have been largely consumed for viral protein synthesis, while simultaneous activation of cellular integrated stress response (ISR) may lead to selective increase in aspartate and asparagine[16]. Interestingly, the small molecule ISR inhibitor ISRIB[17] increased the number of live cells at 48 hpi, even though it did not appreciably diminish viral gRNA or Np expression (Supplementary Fig. 3).

One of the most striking changes in intracellular metabolomics was accumulation of de novo purine synthesis intermediates, including 5-phosphoribosyl-1-pyrophosphate (PRPP), *N*-formylglycinamide ribonucleotide (FGAR), aminoimidazole ribonucleotide (AIR), and succinylaminoimidazolecarboxamide ribonucleotide (SAICAR) in virus-infected cells (Fig. 1f, g, Supplementary Fig. 4, and Supplementary Data 3). Ribonucleotide synthesis requires ribose derived from glucose as well as one-carbon (1C) units carried by folate species. Intracellular glucose and folate were significantly depleted in SARS-CoV-2-infected cells, raising the possibility that host glucose and folate metabolism were hijacked to meet the demand for viral subgenomic RNA replication. Interestingly, this occurs concomitantly with a significant decrease in host mRNA abundance (Fig. 1d), which we speculate were destabilized by the host 'shut-off' activity of SARS-CoV-2 to salvage host nucleotide supply for viral biosynthesis[18]. Of note, inosine monophosphate (IMP) was not significantly increased (Fig. 1g), suggestive of a balance between production and consumption at this regulated step in purine synthesis. Consistent with the observation that SARS-CoV-2 also shuts off translation of most host proteins[8], our results suggest that it evolved a strategy to upregulate purine metabolism on the post-translational level.

Intracellular glucose levels were lower and lactate levels higher at this early timepoint, suggesting increased glycolysis. Glycolysis provides ATP as well as building blocks for de novo serine and nucleotide synthesis. Shifting the media sugar source from glucose to galactose just prior to infection, which allows continued operation of OXPHOS but not glycolysis, strongly impaired production of viral +sense genomic RNA (gRNA), nucleoprotein subgenomic RNA, and infectious virion. Shift to galactose also increased live cell number at 48 hpi, suggesting reduction of viral cytopathic effect (CPE; Fig. 2a, b and Supplementary Fig 5a). Similar results were observed in human lung carcinoma A549 ACE2 + cells (Supplementary Fig. 5b). Treatment of cells with the

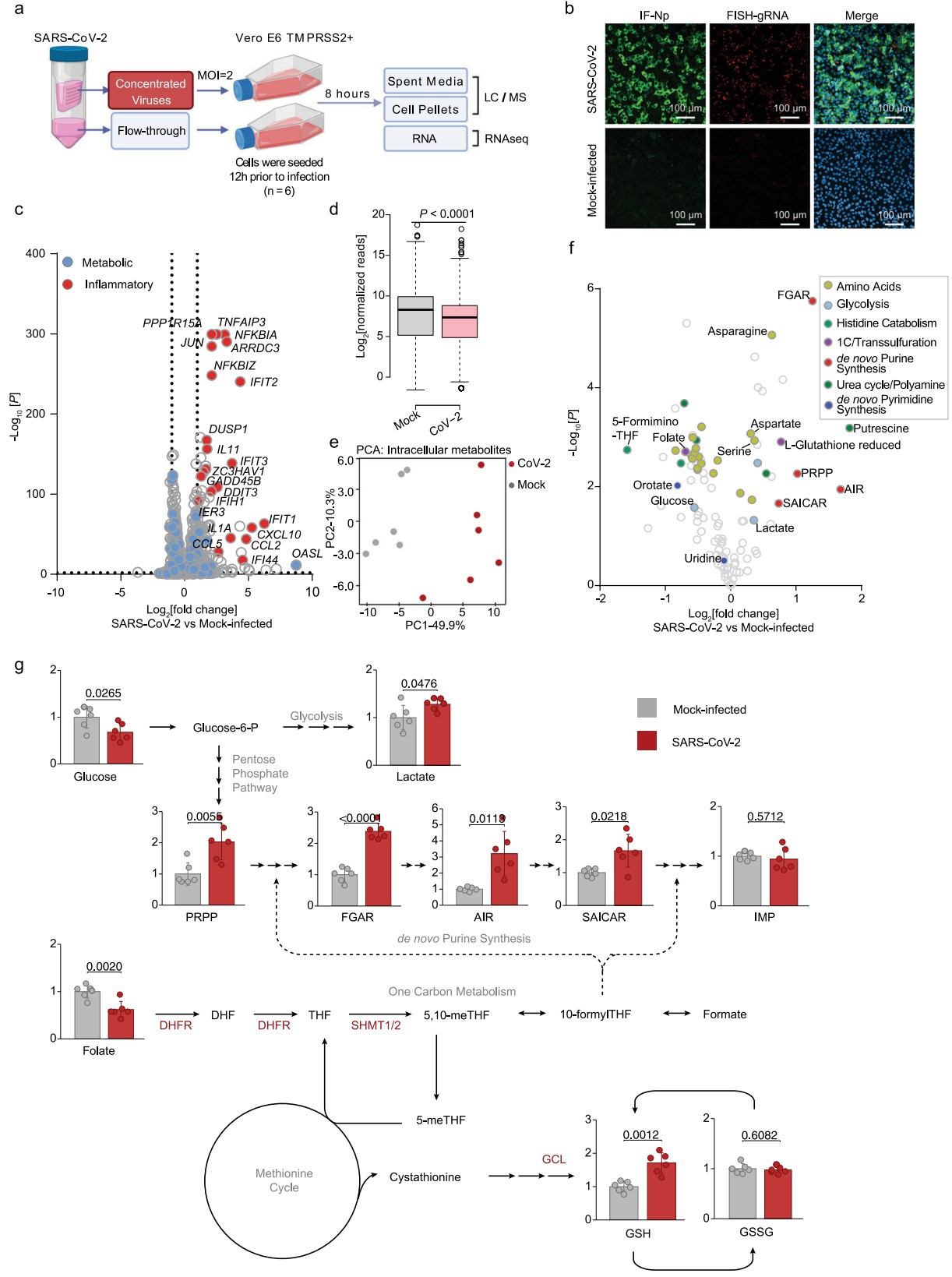

electron transport chain complex I inhibitor Piericidin A (PierA), which blunts bioenergetics by blocking OXPHOS, also reduced viral CPE, but did not have a major impact on viral gRNA, Np, or secreted virus levels (Fig. 2c–e). Taken together, these results suggest that impairment of glucose-derived precursors for anabolic reactions likely underlies the observed decrease in viral replication upon glucose starvation.

Folate metabolism is critical for transfer of 1C units for nucleotide synthesis and is also closely linked to glutathione biosynthesis via the transsulfuration pathway (Supplementary

**Fig. 1 Metabolic changes induced by early SARS-CoV-2 infection. a** Schematic of the metabolic profiling approach. Vero E6 TMPRSS2 + cells were infected with concentrated SARS-CoV-2 at a MOI = 2 or mock-infected with virus-depleted flow-through, infected for 8 h and profiled by LC/MS and RNAseq in parallel. **b** Immunofluorescence of nucleoprotein (IF-Np) and fluorescence in situ hybridization (FISH) for + strand viral genomic RNA (FISH-gRNA) and merge with Hoechst stained nuclei in infected vs mock-infected cells. See also Supplementary Fig. 1b. The experiment was reproduced in at least six independent experiments. **c** Volcano plot visualization of 8 h SARS-CoV-2 versus mock vs infected Vero E6 RNAseq from $n = 3$ datasets. Selected inflammatory (red) and metabolism (blue) pathway genes are shown. $P$-value generated with DESeq under default setting. **d** Box plot visualization of RNAseq reads in SARS-CoV-2 versus mock-infected cells. $n = 3$ biologically independent samples were examined over one independent experiment. Data are presented as mean values ± SD. One-way ANOVA with multiple comparison using the Sidak method. **e** PCA of 106 intracellular metabolites, as determined by LC-MS in SARS-CoV-2-infected (red) or mock-infected (gray) cells, from $n = 6$ biologically independent replicates. **f** Volcano plot visualization of log2 fold change (x-axis) and -log10(P value; y-axis) of intracellular metabolites measured by LC-MS. Significantly increased or decreased metabolites related to glycolysis, de novo purine synthesis, 1C metabolism/ transsulfuration pathway, amino acids, histidine catabolism, urea cycle/ polyamine metabolism, and de novo pyrimidine synthesis are labeled. $n = 6$ biologically independent samples were examined over one independent experiment, $P$-values were generated with two-tailed $P$ value from Student's $t$ test. **g** Fold change of intracellular glucose, lactate, de novo purine, and one-carbon metabolite levels detected by LC-MS in SARS-CoV-2 and mock-infected cells. Mock-infected levels were set to 1. All barplots show mean ± standard deviation (SD). *$P < 0.05$, **$P < 0.01$, or ***$P < 0.001$ from Student's two-tailed $t$ test. Druggable targets are labeled in red. See also Supplementary Fig. 4. Source data are provided as a Source Data file.

Fig. 4)[19]. Depletion of folate in SARS-CoV-2-infected cells was coupled to increased glutathione abundance, raising the possibility that 1C metabolism might support SARS-CoV-2 replication through roles in antioxidant defense (Fig. 1f). However, the gamma-glutamylcysteine synthetase inhibitor L-buthionine-sulfoximine (BSO), which blocks synthesis of glutathione (GSH, GSSG, Fig. 1g), did not have any apparent effect on viral gRNA and Np levels (Supplementary Fig. 5c, d). This result suggests that the increase in SARS-CoV-2-infected cell glutathione pool may be an epiphenomenon of metabolic rewiring and not directly required for viral replication, or may take on a role in microenvironments in vivo. Collectively, these observations raise the hypothesis that SARS-CoV-2 activates glucose and folate metabolism at the post-transcriptional level in newly infected cells to supply the massive need for ribonucleotide synthesis, in a manner that bypasses viral impairment of host mRNA translation.

Guided by these early post-infection metabolomic changes, we asked whether inhibition of folate metabolism can blunt viral RNA expression and virion production. The widely used drug methotrexate (MTX), a folate analog which competitively inhibits the enzyme dihydrofolate reductase (DHFR) as well as several additional steps in 1C metabolism and nucleotide synthesis[20–22], significantly blocked virus-induced CPE, reduced viral gRNA and Np levels, and diminished secretion of infectious virion by nearly 2-log (Fig. 2f–i). All of the mentioned effects of 1 µM MTX could be at least partially reversed by 30 µM hypoxanthine, an intermediate of purine salvage pathway and an alternate source of purines[23]. Interestingly, addition of 100 µM of the pyrimidine deoxynucleotide thymidine or 1 mM of the one-carbon group donor formate could only partially restore viral gRNA levels, but were unable to rescue translation of viral Np, virion production, or CPE (Fig. 2f–i and Supplementary Fig. 6a, b). Similar results were obtained in A549 ACE2 + cells, where MTX diminished gRNA and Np subgenomic RNA expression, viral load, and CPE. These phenotypes were significantly rescued by the addition of hypoxanthine, but not by formate. Thymidine supplementation likewise increased gRNA, but not Np subgenomic RNA expression in A594 ACE2 + cells (Fig. 2j, k). By quantitative RNA Flow-FISH, only hypoxanthine significantly rescued Np subgenomic RNA expression in A549 ACE2 + (Fig. 2l, m). These results suggest that the sensitivity of viral replication to methotrexate is closely related to the critical role of folate metabolism in supporting de novo purine synthesis and are indicative of conserved roles in a physiologically relevant human cell type.

To further isolate the roles of folate species generation and interconversion in viral replication, we inhibited cytosolic and mitochondrial isoforms of serine hydroxymethyltransferase

(SHMT1 and SHMT2, respectively; Fig. 1g)[19]. Treatment of Vero E6 cells just prior to infection with the highly-specific SHMT1/2 dual inhibitor SHIN1[24] reduced infectious virus titer by ~1-log at 48 hpi, diminished + strand gRNA and nucleoprotein levels, and induced resistance to viral CPE (Fig. 3a–c). SHIN1 effects could be rescued by the addition of formate, demonstrating an on-target mechanism of action of the inhibitor (Fig. 3a–c and Supplementary Fig. 7). SHIN1 exerted similar effects in A549 ACE2 + cells on SARS-CoV-2 replication and CPE (Fig. 3d, e), and RNA Flow-FISH demonstrated that SHIN1 also significantly diminished Np RNA expression (Fig. 3f, g). As SHIN1 is a dual SHMT1/2 antagonist, we next used CRISPR to test whether SARS-CoV-2 replication was specifically dependent on a SHMT1- or SHMT2-mediated one-carbon metabolism pathway. CRISPR SHMT1 targeting reduced expression of +strand gRNA, Np RNA and protein, diminished infectious virus titer by nearly 1-log and enhanced cell survival (Fig. 3h, j). By contrast, CRISPR SHMT2 targeting had minimal effect on these parameters of viral infection (Fig. 3h, j). Although we note that there is some residual SHMT2 expression in this experiment, treatment with PierA is also known to indirectly block the mitochondrial 1C pathway[23], but also did not reduce viral replication (Fig. 2e). Taken together, these results suggest that the cytosolic branch of host 1C metabolism is particularly important for virion production, potentially at the level of viral subgenomic RNA expression (Fig. 4).

These results raise the question of whether de novo pyrimidine nucleotide synthesis plays a similarly important role in SARS-CoV-2 replication. Vero E6 were therefore treated with vehicle control or with brequinar, a potent and specific antagonist of the rate-limiting enzyme in de novo pyrimidine biosynthesis dihydroorotate dehydrogenase (DHODH) in uridine-free media to block pyrimidine salvage (Supplementary Fig. 8). Brequinar significantly diminished viral CPE and Np sgRNA expression and reduced viral load by ~1 log, suggesting the importance of de novo pyrimidine synthesis in supporting SARS-CoV-2 replication under uridine-limiting conditions. However, its effects were much milder than those of methotrexate. Curiously, when tested in combination, brequanir seemed to antagonize the effects of methotrexate, though we did not pursue this observation.

Our studies highlight an interesting difference between SARS-CoV-2-infected epithelial cell and monocytes, where glycolytic flux has also been reported to support viral replication[25]. However, in monocytes, SARS-CoV-2 replication was reported to increase mitochondrial reactive oxygen species to trigger a hypoxia-inducible factor-1a-dependent pathway that upregulates glycolysis genes at the transcriptional level at 24 h post-infection[26]. It will therefore be of interest to determine whether

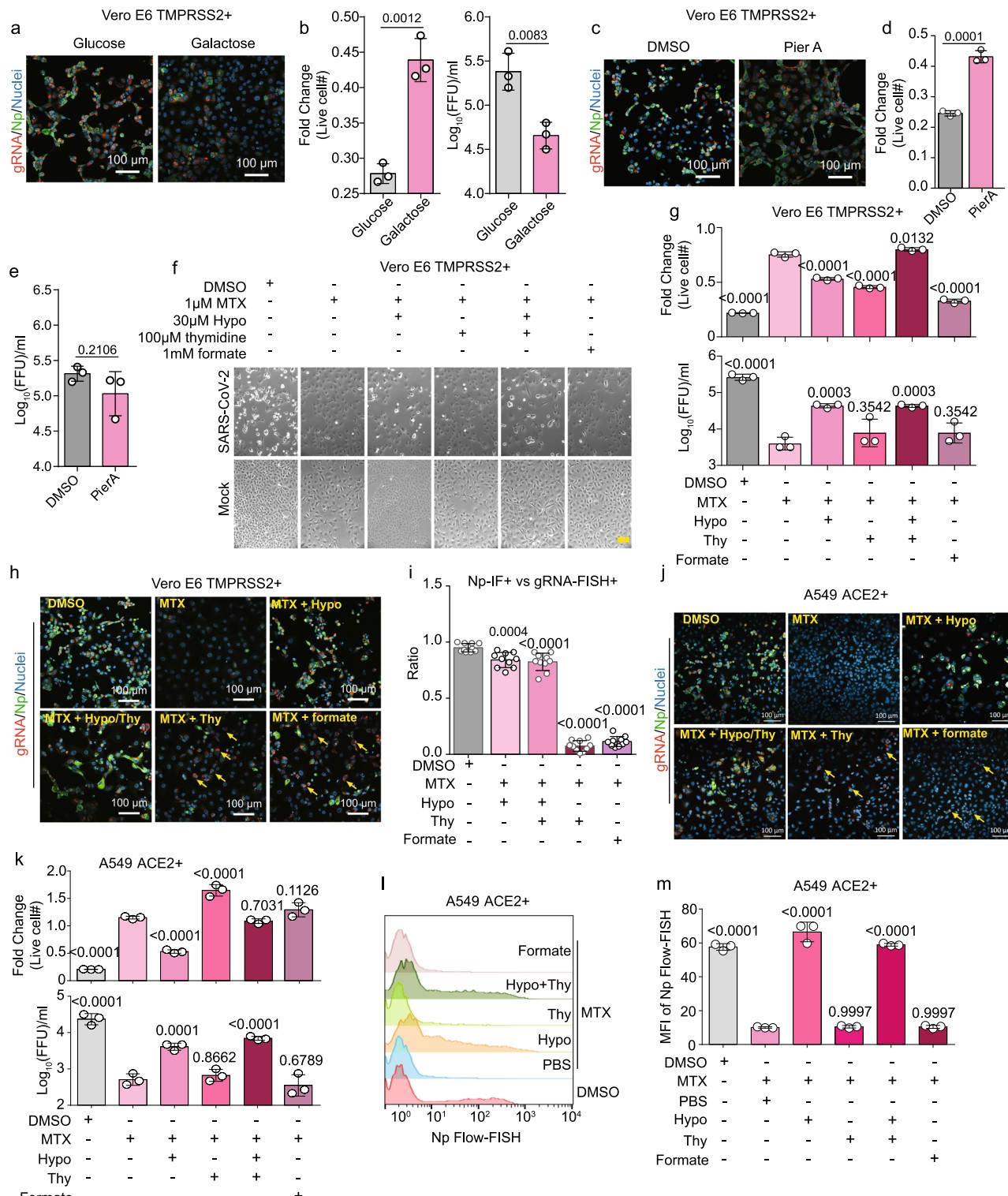

monocyte SARS-CoV-2 replication is dependent on glucose for de novo nucleotide synthesis at earlier timepoints of infection[27].

While previous studies of host/virus metabolism have focused on later timepoints of infection[28], the current analyses point to an unappreciated requirement for host one-carbon metabolism early in the viral lifecycle. How SARS-CoV-2 induces host cell nucleotide metabolism remains to be characterized. Interestingly, viral NSP14 interacts with the host enzyme IMPDH2[29], which catalyzes inosine 5′-phosphate conversion to xanthosine 5′-phosphate in the first

committed and rate-limiting step of de novo guanine synthesis. IMPDH inhibition reduces viral replication[30,31], raising the possibility that NSP14 may activate de novo purine synthesis to support massive viral RNA production. Global phosphoproteomic analysis has also revealed that host kinase signaling is altered at these early timepoints of Vero-E6 SARS-CoV-2 infection, including in pathways such as AKT that can activate metabolism responses[27]. Future studies will be required to distinguish between these and other mechanisms.

**Fig. 2 SARS-CoV-2 induced glycolysis and one-carbon metabolism supports viral RNA and protein expression, replication, and cytopathic effect.**
**a** +strand gRNA FISH, Np IF, and merge with Hoechst stained nuclei of Vero-E6 TMPRSS2 + cells cultured in media with 25 mM glucose versus galactose as the sugar source and infected with SARS-CoV-2. **b** Mean ± SD fold change of live Vero E6 TMPRSS2 + cell number and median tissue culture infectious dose (TCID50) presented as fluorescent-focus units (FFU) per ml of culture supernatant at 48 h post infection of cells cultured in glucose versus galactose from $n = 3$ biologically independent replicates. **c** FISH analysis of +strand gRNA, IF for Np, and merge with Hoechst stained nuclei in SARS-CoV-2-infected Vero E6 TMPRSS2 + cells treated with DMSO or 100 nM piericidin A (PierA). **d** Mean ± SD fold change live cell number from $n = 3$ biologically independent replicates, as in **c**. **e** Mean ± TCID50 from $n = 3$ biologically independent replicates in Vero E6 TMPRSS2 +, as in **b**. **f** Phase microscopic images of SARS-CoV-2 versus mock-infected Vero E6 TMPRSS + cells cultured for 48 h with DMSO, 1 μM of methotrexate (MTX), 30 μM hypoxanthine (hypo), 100 μM thymidine, or 1 mM formate, as indicated. Yellow scale bar indicates 100 μm. **g** Mean ± fold change live cell # and TCID50/ml from samples collected as in **f** from three biologically independent replicates. **h** FISH microscopic analysis of viral gRNA, IF of Np, and merge with Hoechst stained nuclei in SARS-CoV-2-infected Vero E6 TMPRSS2 + cells treated for 48 h with the indicated conditions. Yellow arrows indicate representative cells with high gRNA (red) but low Np (green) signal. **i** Ratios of +strand gRNA FISH versus Np IF signals from 500 Vero E6 TMPRSS2 + cells from 20 random fields for each condition in **h** are shown. **j** FISH microscopic analysis of viral gRNA, IF of Np, and merge with Hoecshst stained nuclei in SARS-CoV-2-infected A549 ACE2 + cells treated with the indicated conditions. Yellow arrows indicate representative cells with high gRNA (red) but low Np (green) signal. **k** Fold change mean ± SD live cell # and TCID50/ml from A549 ACE2 + samples collected as in **j** from $n = 3$ biologically independent replicates. **l** Flow-FISH analysis of Np subgenomic RNA in SARS-CoV-2-infected A549 ACE2 + cells treated with the indicated conditions. Of note, the leftmost peak in each row indicates uninfected cells. **m** Mean ± SD of Np subgenomic RNA mean fluorescence intensity (MFI) values from $n = 3$ biologically independent replicates, as in **l**. In all panels, cells were infected at MOI = 0.1 for 48 h. Microscopy images are representative of at least $n = 3$ biologically independent values. *P*-values in this figure were calculated by one-way ANOVA with multiple comparisons using Sidak method. Source data are provided as a Source Data file.

High levels of transcription enable massive production of coronavirus structural proteins, in particular Np[6,27]. These results, taken together with our studies, are consistent with a model in which de novo purine synthesis is particularly important for vast sgRNA production within the first 8 h of SARS-CoV-2 infection[6] (Fig. 4). Our results support further investigation of antifolates, including methotrexate and SHMT inhibitors[32], as host-directed antivirals against SARS-CoV-2. Taken together with a recent pre-print[33], further investigation of antifolate roles in COVID-19 prophylaxis versus treatment approaches would be of significant interest. Indeed, high-dose methotrexate has also been investigated in treatment of SARS-CoV-2 associated hyper-inflammatory syndromes[34], suggesting that antifolates may be useful in counteracting COVID-19 via two distinct mechanisms: antiviral activity and anti-inflammatory action. Methotrexate effects on the cellular purine nucleotide pool likewise impair replication of the RNA flavivirus Zika[35]. Our work raises important clinical hypotheses that may be rapidly and practically actionable, given that long-term methotrexate therapy is already FDA approved for inflammatory disorders such as rheumatoid arthritis and psoriasis, and that there is ample precedent for modulating host one-carbon pools via dietary modulation[36,37]. Methotrexate or SHMT inhibition might also prove to be synergistic with the antiviral nucleotide analog remdesivir, which competes with ATP for incorporation by the viral RNA polymerase[38,39], and has recently been approved for hospitalized COVID-19.

It is estimated that there may be thousands of coronaviruses in bat reservoirs with potential for human transmission. Given that there have been three recent spillovers of pathogenic coronavirus into the human population during the past two decades, there is great need for preparedness for a future coronavirus outbreak. As it is difficult to predict which strain might next emerge, host-targeted antiviral therapy offers the promise of conferring broad protection against future zoonotic coronaviruses with pandemic potential.

## Methods

**Cells and viruses.** The African Green Monkey Vero E6 with stable TMPRSS2 expression was constructed by lentiviral transduction of Vero E6 (ATCC) using the construct pTRIP-SFFV-Hygro-2A-TMPRSS2 and selection at 500 μg/ml hygromycin. Vero E6 TMPRSS2 + cells were maintained in Dulbecco's Modified Eagle's Medium (DMEM, 25 mM glucose, 4 mM glutamine, 1 mM sodium pyruvate,

Gibco, #11965118) supplemented with 10% fetal bovine serum (FBS), 100 U/mL penicillin/streptomycin and 250 μg/ml hygromycin at 37 °C with 5% CO₂. Cell lines with stable *Streptococcus pyogenes* Cas9 expression were generated by lentiviral transduction using pXPR_BRD111 (Addgene plasmid # 78166, a gift from Dr. William Hahn) and blasticidin selection (5 ug/ml). A549 ACE2 + cells were constructed as follows. A549 cells obtained from ATCC and maintained in Roswell Park Memorial Institute (RPMI) 1640 Medium (Gibco, #11875) supplemented with 10% FBS at 37 °C with 5% CO₂. A549 ACE2 + cells were constructed as follows. An entry vector containing human ACE2 cDNA from the Ultimate ORF library (ThermoFisher Scientific) was Gateway cloned in pHAGE-EF1-DEST using LR Clonase II (ThermoFisher Scientific) per the manufacturer's protocol to make pHAGE-EF1-ACE2. pHAGE-EF1-ACE2 was packaged into lentivirus using the psPAX2 and VSV-G lentiviral packaging system. A549 was transduced with pHAGE-EF1-ACE2 lentivirus. In all, $1 \times 10^7$ A549 ACE2 + cells were washed with phosphate-buffered saline (PBS), stained with 0.25 μg/10⁶ cells of anti-ACE2 antibody (R&D Systems, AF933) in PBS containing 1% BSA for one hour, washed again with PBS, stained with FITC-conjugated anti-goat secondary antibody (Invitrogen, A21467, 1:1000) for 30 min, and washed twice with PBS. ACE2 + cells were sorted using an MA900 multi-application cell sorter (Sony).

All cell lines were routinely tested and certified as mycoplasma-free using the MycoAlert kit (Lonza, LT07-318). SARS-CoV-2 isolate from USA-WA1/2020 (GenBank accession # MN985325) was obtained from BEI Resources (#NR-52281) and worked with exclusively in the Broad Institute BSL-3 laboratory with approval from the Broad Environmental Health and Safety Office. The virus was propagated as described previously[40]. Vero E6 cells were used for virus propagation and titration. For virus stock preparation, 100 μL of passage 0 (P0) virus was mixed with 5 ml of trypsinized cell suspension ($0.25 \times 10^6$/ml) and seeded into a T25 flask. The cell culture supernatant was harvested at 3 dpi and used as the P1 virus stock. We then mixed 100 μl of P1 virus with 10 ml of Vero cells suspension ($0.25 \times 10^6$/ml) and seeded the mixture into a T75 flask. The cell culture supernatant was again harvested at 3 dpi and used as the P2 virus stock for all the experiments in this study. We used same procedure to grow a large amount of P2 viruses, which was carefully tittered with a fluorescent-focus assay.

To acquire highly concentrated virus stock, the viral culture supernatant was concentrated up to ×25 by using the Microsep Advance Centrifugal Devices with Omega Membrane 30K (Pall Corporation, MAP030C38). The flow-through of the centrifugal devices was collected as the mock infection control. Viral stock titer was performed with a fluorescent-focus assay. For CRISPR editing, sequences of sgRNAs against African Green Monkey SHMT1 and SHMT2 listed in Table S4 were cloned into pLentiguide Puro (Addgene # 52963, a gift from Dr. Feng Zhang). Transduced Vero E6 TMPRSS2 cells were selected with puromycin 3 μg/ml at 48 h post-transduction for 5 days.

**Fluorescent-focus assay.** Virus titer in the cell culture supernatant was determined by a fluorescent-focus assay[41]. Briefly, we aliquoted 90 μL of serum-free DMEM into columns 1–12 of a 96-well tissue culture plate, then pipetted 10 μL of viruses supernatant into column 1 and serially diluted 10-fold across the plate. We then trypsinized and resuspended Vero cells in DMEM containing 10% FBS, at a density of $5 \times 10^5$ cells/mL. We added 50 μL of cell suspension directly to each well and mixed gently by pipetting. We then inoculated cultures in a 37 °C incubator with 5% CO₂ for 2 days. The titration plates were fixed with 4% paraformaldehyde (PFA) overnight. Fixed cells were stained with SARS-CoV-2 N protein-specific

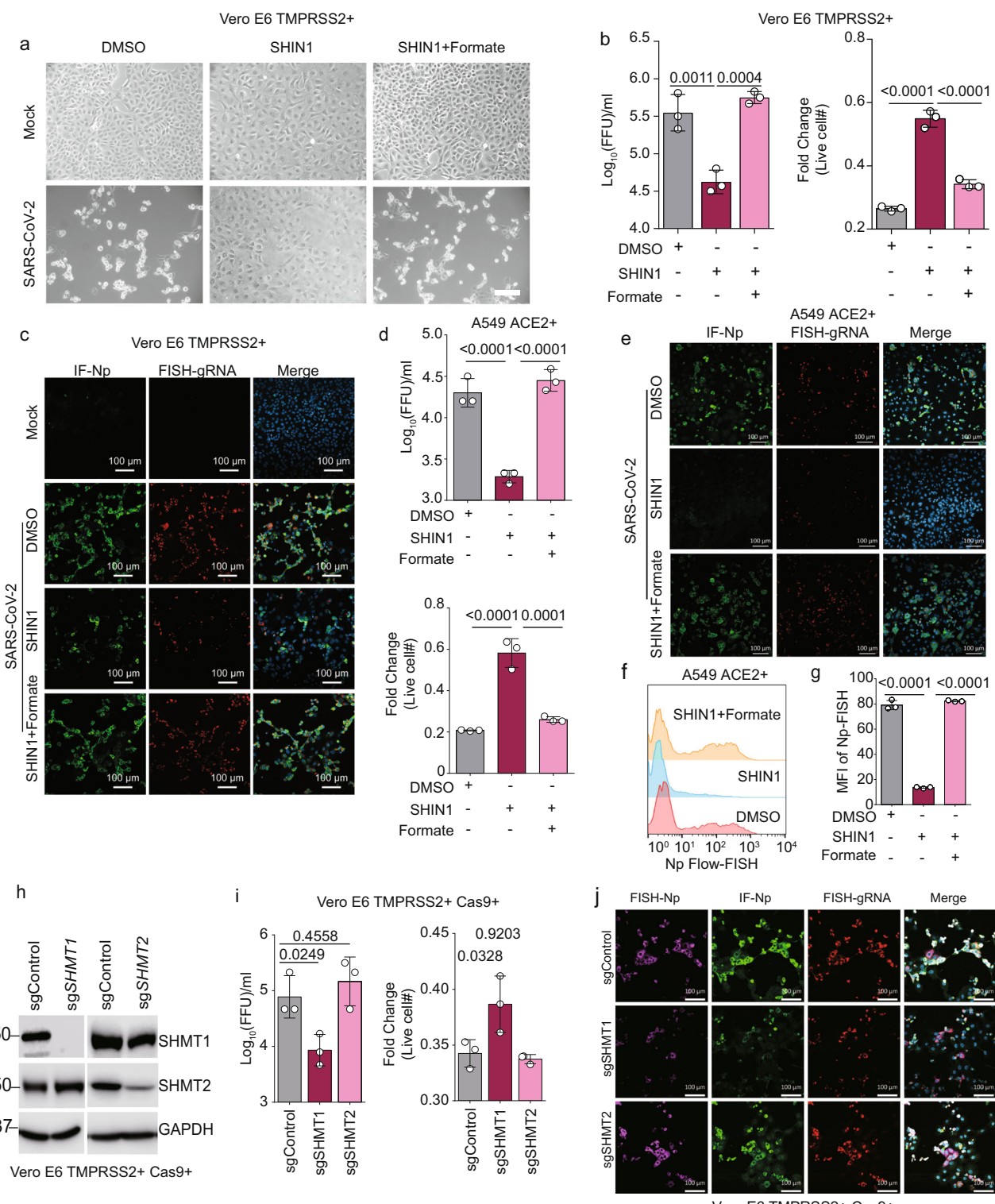

monoclonal antibody SA-46 (MAb) and Alexa Fluor 488-conjugated goat anti-mouse IgG IgG was used as a secondary antibody. Fluorescent foci of infected cells were observed and counted using a phase-contrast fluorescence microscope. Virus titers were presented in number of fluorescent-focus units per ml (FFU/ml).

**Inhibitors and special media conditions**. The small molecules methotrexate (Selleckchem, S1210), hypoxanthine (Sigma, H9636), thymidine (Sigma, T1895), and sodium formate (Fisher Scientific, S648-500) were used. The dual SHMT1/2 inhibitor SHIN1 (Tocris, 6998) was used at the concentration of 10 μM. In total, 1 mM of sodium formate was used to demonstrate on-target effects of SHIN1. Glutamylcysteine synthetase inhibitor ʟ-buthionine-sulfoximine (BSO; Sigma,

83730-53-4) was used at the concentration of 10 μM. Mitochondrial complex I inhibitor Piericidin A (Cayman, 15379) was used at the concentration of 0.1 μM. Brequinar (Cayman, 24445) was used at the concentration of 1 μm and 10 μM. Glucose-free media containing galactose was prepared by supplementing 25 mM galactose (Sigma, G5388) into glucose-free DMEM (#11966025, ThermoFisher) with 10% dialyzed FBS (#26400044, ThermoFisher). Cells were always treated with small molecules or special media 2 h prior to infection. Samples were harvested at 48 hpi.

**Simultaneous Stellaris FISH and immunofluorescence**. Two sets of Stellaris FISH probes, targeting either SARS-CoV-2 ORF pp1a positive strand genomic

**Fig. 3 SARS-CoV-2 induced serine one-carbon metabolism supports viral RNA and protein expression, replication, and cytopathic effect. a** Phase microscopic images of SARS-CoV-2 versus mock infected Vero E6 TMPRSS2 + cells cultured with DMSO, 10 μM of the dual SHMT1/2 inhibitor SHIN1 or 10 μM SHIN1 + 1 μM formate, as indicated. White scale bar indicates 100 μm. The experiment was reproduced in at least six independent experiments. **b** Mean ± SD fold change TCID50 (left) and live cell (right) from $n = 3$ biologically independent replicates, as in **a. c** IF of Np, FISH for +strand gRNA, and merge with Hoechst stained nuclei in mock-infected or SARS-CoV-2-infected Vero E6 TMPRSS2 + cells treated with DMSO, SHIN1, or SHIN1 + formate. **d** Mean ± SD fold change TCID50 (top) and live cell (bottom) values in SARS-CoV-2-infected A549 ACE2 + cells, treated with the indicated conditions, from $n = 3$ biologically independent replicates. **e** IF of Np, FISH for +strand gRNA, and merge with Hoechst stained nuclei in mock-infected or SARS-CoV-2-infected A549 ACE2 + cells treated with DMSO, SHIN1, or SHIN1 + formate. **f** Flow-FISH analysis of Np subgenomic RNA in SARS-CoV-2-infected A549 ACE2 + cells treated with the indicated conditions. **g** Mean ± SD values from $n = 3$ biologically independent replicates of viral subgenomic RNA Flow-FISH MFI values, as in **f. h** Immunoblot analysis of whole cell lysates from Cas9 + TMPRSS2 + Vero E6 expressing control, SHMT1 or SHMT2 sgRNAs. **i** Mean ± SD fold change TCID50 (left) and live cell (right) values from Vero E6 TMPRSS2 + with control, SHMT1 or SHMT2 targeting sgRNAs infected by SARS-CoV-2 from $n = 3$ biologically independent replicates are shown. **j** FISH of subgenomic Np RNA, IF of Np, FISH for +strand gRNA, and merge with Hoechst stained nuclei in cells with control, SHMT1 or SHMT2 targeting sgRNAs infected by SARS-CoV-2. In all panels, cells were infected at MOI = 0.1 for 48 h. Microscopy images are representative of at least $n = 3$ biologically independent values. $P$-values in this figure were calculated by one-way ANOVA with multiple comparisons using Sidak method. Source data are provided as a Source Data file.

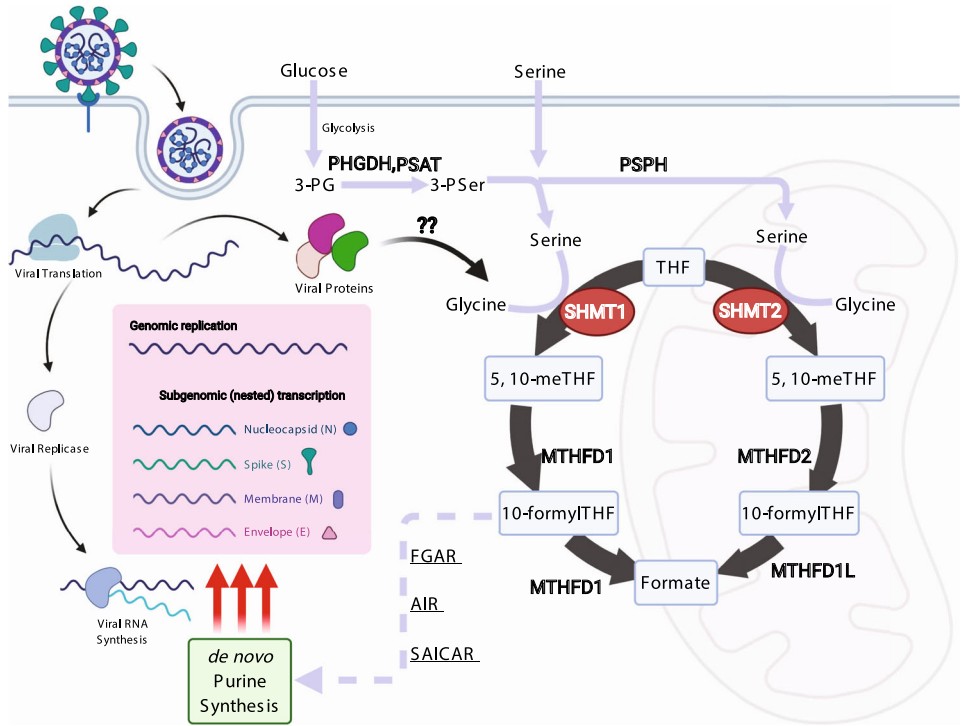

**Fig. 4 Schematic of SARS-CoV-2 induced one-carbon metabolism in support of viral replication.** SARS-CoV-2 induces glycolysis and one-carbon metabolism at the post-transcriptional level in newly infected cells. Serine metabolism, particularly by cytosolic SHMT1 produces carbon units for de novo purine synthesis in support of massive viral subgenomic RNA synthesis, non-structural protein expression, and viral replication.

RNA or the subgenomic RNA encoding ORF9 (nucleocapsid protein) were obtained from Biosearch Technologies (Supplementary Data 4). Nucleocapsid protein was detected by using the mouse monoclonal antibody SA46, kindly provided by Dr. Ying Fang. Cells grown on chambered dishes (Mattek) were fixed with 4% paraformaldehyde (PFA) buffered with PBS overnight in the BSL3 laboratory. Cells were then permeabilized with 70% ethanol for at least 2 h and subjected to FISH and immunofluorescence following the product manual from Stellaris RNA-FISH. Briefly, cells were incubated with primary antibodies against nucleocapsid (the monoclonal antibody SA46, 1:1000), genomic RNA FISH probes (1:1000), or subgenomic RNA FISH probes (1:1000) in Hybridization Buffer (#SMF-HB1-10, Biosearch Technologies) containing 10% formamide for at least 4 h at 37 °C. Slides were then washed three times with Buffer A and then incubated with secondary antibodies (Alexa Fluor 488-conjugated goat antimouse 1:250 in Buffer A, #SMF-WA1-60, Biosearch Technologies) for 1 h at 37 °C. Finally, slides were washed three times in Buffer B (#SMF-WB1-20, Biosearch Technologies) and incubated with 100 μL of Hoechst 33258 staining buffer (10 μg/mL in PBS) for 10 min to stain nuclear DNA. Cells were then washed two times with Buffer B. ProLong Gold anti-fade reagent (P36935, ThermoFisher) was applied to the slide, which was then sealed with a No. 1.5 coverslip. Image acquisition was performed with the Zeiss LSM 800 instrument. Image analysis was performed with the Zeiss ZEN Blue software.

**Np sgRNA Flow-FISH**. Vero E6 TMPRSS2 + cells or A549 ACE2 + cells were seeded in T25 flasks at the density of $0.25 \times 10^6$/ml 12 h prior to infection. Cells were mock infected or infected with SARS-CoV-2 at a MOI of 0.1. At 48 hpi, cells were trypsinized and resuspended in DMEM or RPMI containing 10% FBS. Cells were washed once with PBS and fixed with 4% PFA overnight in the BSL3 laboratory. Cells were then permeabilized with 70% ethanol for 2 h at room temperature and subjected to FISH, following the Stellaris RNA-FISH product manual (LGC Biosearch Technologies). Briefly, cells were washed once by buffer A (SMF-WA1-60, LGC Biosearch Technologies) and then incubated with anti-Np sgRNA FISH probes (1:100) in Hybridization Buffer (#SMF-HB1-10, LGC Biosearch Technologies) containing 10% formamide overnight at 37 °C. Cells were then washed twice in Buffer A and once with PBS. Cells were then sorted on a BD FACSCalibur cytometer and analyzed with FlowJo X software (FlowJo).

**Live cell number (cytopathic effect) quantification assay**. Cell number was quantified by Viral Tox-Glo (VTG, G8943, Promega) live cell assay as described previously[42]. Normalized cytopathic effect was calculated by normalizing VTG values of the infected samples to the VTG values of the corresponding uninfected samples. To normalize the ATP suppressing effects caused by media change, the fold change of live cell number was calculated dividing the luminescence signal of

the SARS-CoV-2-infected cells by the signal of mock infected cells with same treatment.

**Immunoblot analysis**. Immunoblot analysis was performed as previously described[42]. In brief, whole cell lysates (WCL) were separated by SDS-PAGE electrophoresis, transferred onto the nitrocellulose membranes, blocked with 5% milk in TBST buffer and then probed with relevant primary antibodies at 4 °C overnight, followed by secondary antibody (Cell Signaling Technology) incubation for 1 h at room temperature. Blots were then developed by incubation with ECL chemiluminescence for 1 min (Millipore) and images were captured by Licor Fc platform. The SHMT1 (D3B3J), SHMT2 (E7F4Q), and GAPDH (D16H11) polyclonal antibodies were purchased from Cell Signaling Technology.

**RNAseq analysis**. Vero E6 TMPRSS2 + cells were mock infected or infected by SARS-CoV-2 at MOI = 2 for 8 h. Total RNA from mock infected or SARS-CoV-2-infected cells was isolated using TRIzol Reagent (ThermoFisher, 15596026) following the product manual. To construct indexed libraries, 1 μg of total RNA was used for polyA mRNA-selection, using NEBNext Poly(A) mRNA Magnetic Isolation Module (New England Biolabs), followed by library construction via NEBNext Ultra RNA Library Prep Kit for Illumina (New England Biolabs). Each experimental treatment was performed in triplicate. Libraries were multi-indexed, pooled, and sequenced on an Illumina NextSeq 500 sequencer using single-end 75 bp reads (Illunima). Adaptor-trimmed Illumina reads for each individual library were mapped back to the Chlorocebus sabaeus ChlSab1.1 transcriptome assembly using STAR2.5.2b[43]. Feature Counts was used to estimate the number of reads mapped to each contig[44]. Only transcripts with at least five cumulative mapping counts were used in this analysis. DESeq2 was used to evaluate differential expression (DE)[45]. DESeq2 uses a negative binomial distribution to account for overdispersion in transcriptome datasets. It is conservative and uses a heuristic approach to detect outliers while avoiding false positives. Each DE analysis was composed of a pairwise comparison between experimental group and the control group. Differentially expressed genes were identified after a correction for false discovery rate (FDR). For more stringent analyses, we set the cutoff for truly differentially expressed genes as adjusted $p$ value (FDR corrected) <0.05 and absolute fold change >2. The volcano plots were built based on the log2(fold-change) at $x$-axis and −log10 ($P$-value) at $y$-axis with Graphpad Prism7.

**Cell culture for intracellular and media metabolites profiling**. In all, $3.5 \times 10^6$ Vero E6 cells were seeded in T25 flask with 10 mL of fresh media (DMEM, 25 mM glucose, 4 mM glutamine, 1 mM sodium pyruvate, Gibco, #11965118) supplemented with 10% fetal bovine serum (FBS), and 100 U/mL penicillin/streptomycin. Twelve hours after seeding, cells were infected with either 500 μL of concentrated SARS-CoV-2 (MOI = 2) virus or 500 μL of virus-depleted flow-through. Media controls without cells were maintained in parallel throughout the experiment for quantitative media consumption and production analysis.

**Intracellular metabolite profiling**. At 8 hpi, media was collected for quantitative metabolite consumption and production analyses, as described below. After removing all media and washing cells with 5 mL of room temperature PBS, 1 mL of dry ice-cold 80% methanol was added to cell monolayer to quench metabolism. Cells were incubated at −80 °C for 30 min, harvested with cell scraper, and centrifuged at $21,000 \times g$ for 5 mins to precipitate proteins. The supernatant was collected in pre-chilled tubes and stored at −80 °C. On the day of analysis, the supernatant was incubated on ice for 20 min and centrifuged at $21,000 \times g$ at 4 °C to clarify. The supernatant was dried down in a speed vacuum concentrator (Savant SPD 1010, Thermofisher Scientific) and resuspended in 100 μL of 60/40 acetonitrile/water. The samples were then vortexed, sonicated in ice-cold water for 1 min, and incubated on ice for 20 mins. Supernatant was collected in an autosampler vial after centrifugation at $21,000 \times g$ for 20 min at 4 °C. Pooled QC samples were generated by combining ~15 μL of each sample. Metabolite profiling was performed using Dionex Ultimate 3000 UHPLC system coupled to Q-Exactive plus orbitrap mass spectrometer (ThermoFisher Scientific, Waltham, MA) with an Ion Max source and HESI II probe operating in switch polarity mode. Zwitterionic Sequent zic philic column (150 × 2.1 mm, 5 μm polymer, part # 150460, MilliporeSigma, Burlington, MA) was used for polar metabolite separation. Mobile phase A (MPA) was 20 mM ammonium carbonate in water, pH 9.6 (adjusted with ammonium hydroxide), and MPB was acetonitrile. The column was held at 27 °C, injection volume 5 μL, autosampler temperature 4 °C, and LC conditions at flow rate of 0.15 mL/min were: 0 min: 80% B, 0.5 min: 80% B, 20.5 min: 20% B, 21.3 min: 20%B, and 21.5 min: 80% B with 7.5 min of column equilibration time. MS parameters were: sheath gas flow 30, aux gas flow 7, sweep gas flow 2, spray voltage 2.80 kV for negative and 3.80 kV for positive, capillary temperature 310 °C, S-lens RF level 50, and aux gas heater temp 370 °C. Data acquisition was done using Xcalibur 4.1 (ThermoFisher Scientific) and performed in full scan mode with a range of 70–1000 $m/z$, resolution 70,000, AGC target 1e6, and maximum injection time of 80 ms. Data analysis was performed in Compound Discoverer 3.1 and Tracefinder 4.1. Samples were injected in randomized order and pooled QC samples were injected regularly throughout the analytical batch. Metabolite annotation was done base on accurate mass (±5 ppm) and matching retention time

(±0.5 min) as well as MS/MS fragmentation pattern from the pooled QC samples against in-house retention time +MSMS library of reference chemical standards. Metabolites with CV < 30% in pooled QC were used for the statistical analysis. The quality of integration for each metabolite peak was reviewed. Polyamines and sulfur-containing metabolites were detected using the method described in quantitative media analysis.

**Media quantitative analysis**. At 8 hpi, spent media was collected, centrifuged at $300 \times g$ for 4 min and kept on ice. In all, 30 μL of spent media was extracted with 120 μL of ice-cold acetonitrile containing metabolomics amino acid mix standard from Cambridge Isotope (MSK-A2-1.2), $^{13}C_6$ -glucose, $^{13}C_3$ -pyruvate, $^{13}C_3$ -lactate, $^{13}C_5$ -glutamine, and $^{13}C_4$ -pyridoxine as internal standards and incubated on ice for 20 mins to precipitate proteins. After centrifugation at $21,000 \times g$ for 5 mins, 90 μL of supernatant was collected and stored at −80 °C. On the day of analysis, samples were incubated on ice for 20 min and the supernatant was collected in an autosampler vial after centrifugation at $21,000 \times g$ at 4 °C for 20 min. Calibration curves were prepared in water at varying concentrations depending on the amino acid/metabolite level in DMEM media. Metabolite separation was done using XBridge amide (2.1 × 100 mm, 2.5 μm, part # 186006091, Waters Corporation, MA). Mobile phase A was 90/5/5 water/acetonitrile/methanol, 20 mM ammonium acetate, 0.2% acetic acid, and mobile phase B was 90/10 acetonitrile/water, 10 mM ammonium acetate, and 0.2% acetic acid. The column temperature was 40 °C, injection volume 5 μL and the flow rate was 0.3 mL/min. The chromatographic gradient was 0 min: 95% B, 5 min: 70% B, 5.5 min: 40% B, 6 min: 40% B, 6.5 min: 30% B, 7 min: 30%B, 7.1 min: 20% B, 8.6 min: 20% B, 8.7 min: 95% B, and 12.5 min: 95% B. MS parameters were same as those mentioned in the intracellular metabolite profiling method. Samples were injected in randomized order and pooled QC samples were injected regularly throughout the analytical batch. Data analysis was performed using Tracefinder 4.1 and respective internal standards were used to calculate the absolute concentration. External calibration curve was used for the quantification of asparagine, nicotinamide, and tryptophan.

**Virus inactivation assay**. We tested whether solvents used for cell pellet and spent media metabolomics (dry ice-cold 80% methanol and ice-cold acetonitrile, respectively) inactivated the virus for safe future experimentation. In total, $5 \times 10^4$ /well of Vero E6 cells in a 96-well plate were infected with 10 μL of samples treated with the solvents. Each sample was validated in triplicates. After culturing for 72 h, we confirmed absence of viral protein, indicated by negative signal in nucleocapsid immunofluorescence assay.

**Statistical analysis**. Unless otherwise indicated, all bar graphs represent the arithmetic mean of three independent experiments ($n = 3$), with error bars denoting standard deviations. Data were analyzed using two-tailed paired Student $t$ test or analysis of variance (ANOVA) with the appropriate post-test using GraphPad Prism7 software. $P$ values correlate with symbols as follows: ns = not significant, $p > 0.05$; *$p < 0.05$; **$p < 0.01$; ***$p < 0.001$; ****$p < 0.0001$.

**Graphics**. Figures were drawn with GraphPad, Biorender, and ggplot2 in R.

**Reporting summary**. Further information on research design is available in the Nature Research Reporting Summary linked to this article.

## Data availability

RNAseq results are available in Table S1. Metabolomic results are available in Table S2-3. We have uploaded the RNAseq dataset into the Gene Expression Omnibus (GEO) with accession # GSE161881 [https://www.ncbi.nlm.nih.gov/geo/query/acc.cgi?acc=GSE161881]. Raw mass spectral data files are available from corresponding authors upon request. Source data are provided with this paper.

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

## Acknowledgements

This work was supported by R01 AI137337 and CA228700 and a Burroughs Wellcome Career Award in Medical Sciences to B.E.G., Howard Hughes Medical Institute and NIH R35 GM122455 to V.K.M. and EMBO ALTF 486-2018 to M.G., and by MassCPR support to S.J.E. We thank Nir Hacohen for helpful discussions and reagents, James Gomez, Rhonda O'Keefe, and Aviv Regev for support with BSL-3 work, John Doench for sgRNA sequences against African Green Monkey SHMT1/2, and Craig Wilen for suggestions about CRISPR editing in Vero E6.

## Author contributions

Y.Z., R.G., S.H.K. H.S., D.T.H., V.K.M., and B.E.G. conceptualized the study. Y.Z. and S.Z. performed virus infection studies. Y.Z., R.G., S.H.K., and H.S. performed the metabolomic analyses. R.G performed RNAseq analyses. Y.Z., R.G., and J.H.L performed biochemical and microscopy analyses. Y.F., M.G., C.N.O., and S.J.E. provided expertize and reagents. S.H.K., R.G., and H.S. performed metabolomic analysis. V.K.M. and B.E.G. supervised the study. Y.Z., R.G., S.H.K., H.S., V.K.M., and B.E.G. drafted the manuscript.

## Competing interests

V.K.M. and B.E.G. are listed as inventors on a patent application filed by the Broad Institute based on results from this manuscript. V.K.M. is on the scientific advisory board and receives compensation from Janssen Pharmaceuticals and 5am Ventures and owns equity in Raze Therapeutics. B.E.G. receives research support from Abbvie. S.J.E. is a founder of TSCAN Therapeutics, MAZE Therapeutics, and Mirimus. S.J.E. is a founder of TSCAN Therapeutics, MAZE Therapeutics, ImmuneID, and Mirimus. S.J.E. serves on the scientific advisory board of Homology Medicines, TSCAN Therapeutics, MAZE Therapeutics, XChem, and is an advisor for MPM, none of which affect this work. Y.Z., R.G., S.H.K., H.S., S.Z., J.H.L., Y.F., M.G., C.N.O., and D.T.H. declare no competing interests.
