## [Peer Review File · Nature Communications]

Reviewers' Comments:

Reviewer #1:

Remarks to the Author:

This is an interesting study where Zhang Y et al. investigate the metabolic reprogramming of African Green Monkey Vero E6 cells upon SARS-CoV-2 viral infection.

The authors identified that intracellular levels of several intermediates of de novo purine synthesis such as PRPP, FGAR, AIR and SAICAR were increased upon SARS-CoV-2 infection. Because of depleted intracellular glucose and folate, authors hypothesize that SARS-CoV-2 infections hijacks glucose and folate metabolism to support the de novo purine synthesis and increased viral RNA replication. Later, authors target one-carbon folate metabolic pathway and show that treatment with a folate analog, methotrexate, or with a SHMT1/2 inhibitor, SHIN1, diminished viral infection and increased host cell viability.

The findings are interesting and critical for understanding metabolic changes that occur in host cells upon SARS-CoV-2 infection and suggest a possible therapeutic approach to fight the infection. The study is well done with appropriate controls and would be of interest to the readership of the Nature Communications. In order to improve the manuscript, there are a couple of points that need to be addressed:

Minor points:

- It would be interesting to investigate metabolic changes that occur in pyrimidine synthesis v. purine synthesis upon infection by SARS-CoV-2. Looking at the intracellular metabolites measurements on Table 3, ureidosuccinic acid (or carbamyl-L-aspartate) and L-aspartate are both increased upon infection. L-aspartate through carbamyl-L-aspartate and enzyme CAD contributes to the de novo pyrimidine synthesis. This potentially indicates that de novo pyrimidine synthesis is also increased due to SARS-CoV-2 infection. Authors test adding thymidine upon inhibition of folate metabolism by methotrexate which only moderately rescues viral infection, however, this experiment is done under inhibited folate metabolism and does not rule out the possible importance of pyrimidine synthesis for viral replication. If authors have measured other intermediates of the de novo pyrimidine synthesis with their metabolomics assay, it would be very interesting to investigate changes in the intracellular levels of pyrimidines upon SARS-CoV-2 infection. Additionally, it would be interesting to inhibit the pathway through which L-aspartate fuels pyrimidines by itself and in combination with methotrexate or SHIN1 to test possible synergistic effect of inhibiting both one-carbon folate metabolism and synthesis of pyrimidines through carbamyl-L-aspartate.
- Authors should explain any normalization that they have done on the metabolomics data prior to comparative analysis of mock and infected cells. At least cell number normalization seems necessary.

Reviewer #2:

Remarks to the Author:

The authors describe the transcriptome and metabolome of SARS-CoV-2 infected Vero-E6 cells 8 hours post infection. The manuscript is well written and the data supports the conclusions. They very effectively describe the specific changes of metabolites involved in central carbon metabolism, and show that metabolites derived from glycolysis are essential for viral survival. The authors further show that folate metabolism specifically is essential, as an inhibitor of dihydrofolate reductase (MTX) prevents viral cytopathic effect (CPE), which can be complimented fully by the purine intermediate hypoxanthine, but only partially by the pyrimidine thymidine, or the intermediate formate. The subsequent enzyme SHMT is similarly shown to be required, and inhibitors thereof can be complemented by formate. Considering our limited knowledge regarding the metabolic pathways and pathogenesis of SARS-CoV-2, these findings are of great interest to a broad audience. However, a few

issues should be addressed prior to publications.

The main concern of this reviewer is the fact that these studies are performed in Vero-E6 cells (African green monkey kidney-derived cell line). Although Vero-E6 cells support SARS-CoV-2 replication/propagation, they are starkly different from human cells that are mainly targeted by this virus (PMID: 32698190). It is, therefore, unclear if these findings are relevant in understanding the biology of SARS-CoV-2. The study's impact would be significantly improved if some of the key findings are validated (using inhibitor/s or CRISPR deletion) in physiologically relevant cell types, such as the human lung epithelial cell line Calu-3.

Reviewer #3:

Remarks to the Author:

This manuscript provides global RNAsequencing and metabolomic data of Vero monkey kidney epithelial cells infected with SARS-CoV-2 at the single early time point, 8 hours post infection. While they do not find significant changes to the transcript levels of major metabolites or metabolic pathways, they find significant changes to the nucleotide synthesis pathway. In particular, they show that members of purine synthesis stemming from the pentose phosphate pathway are increased at 8 hours post infection while intracellular glucose levels are decreased. Importantly, this pathway also requires THF which can be added from one carbon metabolism stemming from Folate. They also show that intracellular Folate is decreased in the infected Vero cells, which they propose is being depleted by the need for nucleotide synthesis. While they would require carbon tracing to demonstrate this claim, it is a reasonable hypothesis. They go on to show that Methotrexate, that inhibits DHFR, the first step of metabolizing Folate, leads to decreased virus production and increased infected cell viability. They also show that an inhibitor of a subsequent step, SHIN1 that inhibits SHMT, also decreases virus production and increases cell viability. They then knockout the different SHMTs to show that it is the cytoplasmic enzyme not the mitochondrial enzyme that is required. They propose that MTX, an approved drug in humans, combined with other antivirals could provide a novel treatment for COVID19 patients. Overall, this manuscript is well done and provides important initial results regarding the importance of specific metabolic changes during SARS-CoV-2 infection. While the use of the long-term passaged Vero cells is limiting for some metabolic studies and could limit the identification of some metabolic changes that are induced by SARS-CoV-2 in more relevant primary cells, they have made significant findings in the Vero cells. There are a couple of minor comments to address.

1. In the text they state that "Quantitative production and consumption analyses of spent media metabolites found subtle, mostly non-significant changes between infected and mock-infected cells at this earlytimepoint." However, Extended figure 3 appears to show increases in lactate and a couple of other metabolites and slight decreases in glucose, glutamine and pyruvate. Are these not significant? If they are they could be mentioned in the results section.
2. In figure2A and C and multiple other panels, they use IFA to quantify viral genomic RNA. They do not demonstrate that this is a truly quantitative assay to show viral RNA levels. Could viral RNAs be directly measured using quantitative RT-PCR? Also, they should quantify the percent of infected cells for these experiments since they do not show differences in host transcripts for the metabolic proteins.
3. At the bottom of page 3 they make reference to extended figure 3C but to show the statement made in that sentence they should reference extended figure 3C and 3D

Michael Lagunoff

Reviewer #4:

Remarks to the Author:

Summary:

In "SARS-CoV 2 Hijacks Folate and One-Carbon Metabolism for Viral Replication" Zhang et. al. demonstrate the importance of folate and one carbon metabolism for SARS-CoV 2 infection. The overall goal of the paper was to identify key host metabolic pathways modified by SARS-CoV 2 to allow replication. Overall, the paper is well written and the experimental conclusions are consistent with the data. How human CoV manipulate host metabolic pathways is a timely subject matter and the authors' conclusions will be of interest to a broad audience.

General Points

As there are several SARS-CoV 2 permissive recombinant human lung epithelial cell lines available why were Vero E6 cells chosen for this study?

Specific Points

In Figure 1F it is not always clear which data point the label belongs with. The data point that corresponds with Serine is the most challenging for this reviewer.

This reviewer appreciates the amount of time put into capturing the data and making figure 1G, however it is overwhelming. Maybe this could be supplemental and a summary table provided to go with the text. This is only a suggestion.

Figure 2b there is less than 1 log difference in FFU/mL when comparing glucose and galactose. The image and graph show different levels of impairment. Less than one log would not be "strongly impaired".

Reviewer #1 (Remarks to the Author):

This is an interesting study where Zhang Y et al. investigate the metabolic reprogramming of African Green Monkey Vero E6 cells upon SARS-CoV-2 viral infection.

The authors identified that intracellular levels of several intermediates of de novo purine synthesis such as PRPP, FGAR, AIR and SAICAR were increased upon SARS-CoV-2 infection. Because of depleted intracellular glucose and folate, authors hypothesize that SARS-CoV-2 infections hijacks glucose and folate metabolism to support the de novo purine synthesis and increased viral RNA replication. Later, authors target one-carbon folate metabolic pathway and show that treatment with a folate analog, methotrexate, or with a SHMT1/2 inhibitor, SHIN1, diminished viral infection and increased host cell viability.

The findings are interesting and critical for understanding metabolic changes that occur in host cells upon SARS-CoV-2 infection and suggest a possible therapeutic approach to fight the infection. The study is well done with appropriate controls and would be of interest to the readership of the Nature Communications. In order to improve the manuscript, there are a couple of points that need to be addressed:

>>Thanks for these positive comments and excellent summary

Minor points:

• It would be interesting to investigate metabolic changes that occur in pyrimidine synthesis v. purine synthesis upon infection by SARS-CoV-2. Looking at the intracellular metabolites measurements on Table 3, ureidosuccinic acid (or carbamyl-L-aspartate) and L-aspartate are both increased upon infection. L-aspartate through carbamyl-L-aspartate and enzyme CAD contributes to the de novo pyrimidine synthesis. This potentially indicates that de novo pyrimidine synthesis is also increased due to SARS-CoV-2 infection. Authors test adding thymidine upon inhibition of folate metabolism by methotrexate which only moderately rescues viral infection, however, this experiment is done under inhibited folate metabolism and does not rule out the possible importance of pyrimidine synthesis for viral replication. If authors have measured other intermediates of the de novo pyrimidine synthesis with their metabolomics assay, it would be very interesting to investigate changes in the intracellular levels of pyrimidines upon SARS-CoV-2 infection.

• **Additionally, it would be interesting to inhibit the pathway through which L-aspartate fuels pyrimidines by itself and in combination with methotrexate or SHIN1 to test possible synergistic effect of inhibiting both one-carbon folate metabolism and synthesis of pyrimidines through carbamyl-L-aspartate.**

>>> Thanks for these important points. Our steady state metabolomics provides a snapshot of changes that follow viral infection and can be useful in generating hypotheses. As requested, we re-examined our metabolomics data with a focus on intermediates of *de novo* pyrimidine synthesis (see below). The only intermediates that were reliably detected in our experiment were carbamoyl aspartate (also known as ureidosuccinic acid) and orotate. Carbamoyl aspartate was significantly increased and orotate significantly decreased in SARS-CoV-2 infected cells. However, it is difficult to draw conclusions about the dependence of viral replication on host *de novo* pyrimidine synthesis from steady-state measurements of two pathway intermediates. Interpretation of changes in abundance of relevant amino acids—glutamine, glutamate, and aspartate—is also complicated, given their involvement in so many pathways, including *de novo* purine synthesis. Moreover, the metabolomics experiment was performed on cells that have been growing in media supplemented with FBS, which contains uridine. Since uridine can be used for pyrimidine salvage, this renders *de novo* pyrimidine synthesis non-essential.

For these reasons, we directly addressed this question by testing sensitivity of viral RNA expression, replication and cytopathic effect to brequinar, an inhibitor of dihydroorotate dehydrogenase (DHODH), the rate-limiting enzyme in *de novo* pyrimidine synthesis. Brequinar experiments were performed in cells cultured with uridine-depleted dialyzed FBS (dFBS) to prevent uridine salvage pyrimidine pathway activity. Interestingly, brequinar significantly diminished subgenomic RNA synthesis, viral load and protected cells from cytopathic effect, suggesting that the *de novo* pyrimidine synthesis pathway is also important for SARS-CoV-2 replication under uridine-limiting conditions. As requested, we also tested brequinar in combination with methotrexate. However, brequinar was not additive with methotrexate. These results are shown in new **Extended Figure 8**.

- **Authors should explain any normalization that they have done on the metabolomics data prior to comparative analysis of mock and infected cells. At least cell number normalization seems necessary.**

SARS-CoV-2 infection does not cause its cytopathic effect until 24-48 hours post infection in Vero E6. Moreover, given that the seeding density was consistent across all samples and that cells were infected with concentrated virus for only 8 hours, we assumed that there would be no significant changes in cell number between the six mock and six virus infected samples. To corroborate our assumption, we have now added additional data to new **Extended Figure 2c**, demonstrating the sum of all known metabolite signals (column sum). This analysis serves as a loading control and demonstrates that column sums were not significantly different between mock and virus-infected groups, suggesting that we captured comparable amounts of metabolites from all samples. In our experience as well as others' (PMID: 31289941), if the total integrated sum of all metabolite peaks are comparable across conditions, this offers excellent evidence that cell input is appropriately normalized.

Reviewer #2 (Remarks to the Author):

The authors describe the transcriptome and metabolome of SARS-CoV-2 infected Vero-E6 cells 8 hours post infection. The manuscript is well written and the data supports the conclusions. They very effectively describe the specific changes of metabolites involved in central carbon metabolism, and show that metabolites derived from glycolysis are essential for viral survival. The authors further show that folate metabolism specifically is essential, as an inhibitor of dihydrofolate reductase (MTX) prevents viral cytopathic effect (CPE), which can be complemented fully by the purine intermediate hypoxanthine, but only partially by the pyrimidine thymidine, or the intermediate formate. The subsequent enzyme SHMT is similarly shown to be required, and inhibitors thereof can be complemented by formate. Considering our limited knowledge regarding the metabolic pathways and pathogenesis of SARS-CoV-2, these findings are of great interest to a broad audience. However, a few issues should be addressed prior to publications.

• The main concern of this reviewer is the fact that these studies are performed in Vero-E6 cells (African green monkey kidney-derived cell line). Although Vero-E6 cells support SARS-CoV-2 replication/propagation, they are starkly different from human cells that are mainly targeted by this virus (PMID: 32698190). It is, therefore, unclear if these findings are relevant in understanding the biology of SARS-CoV-2. The study's impact would be significantly improved if some of the key findings are validated (using inhibitor/s or CRISPR deletion) in physiologically relevant cell types, such as the human lung epithelial cell line Calu-3.

>>Thanks for this important point. We agree that confirmation of key findings in a human cell line is important and increases the study impact. We present new data in **Figures 2j-m, 3d-g and Extended Figs. 5C, 6b and 7b**, using the lung epithelial cell line A549, transduced to stably express the human ACE2 receptor (A549 ACE2+). We used A549 instead of Calu-3, as we could achieve a higher percentage of SARS-CoV-2 infection in A549, and since this cell line has been used for transcriptomic profiling by the tenOever lab. We now provide data that SHIN1 and methotrexate each have similar effects in A549 as in Vero E6. Specifically, SHIN1 and methotrexate protect against SARS-CoV2 cytopathic effect (CPE), reduce viral load and diminish subgenomic RNA expression. Methotrexate effects were partially but significantly rescued by hypoxanthine, as in Vero E6. SHIN1 effects are reversed with the addition

of formate, as in Vero E6. Switching glucose to galactose diminished viral load and viral Np subgenomic RNA and protected A549 ACE2+ cells from CPE, as in Vero E6.

Reviewer #3 (Remarks to the Author):

This manuscript provides global RNA sequencing and metabolomic data of Vero monkey kidney epithelial cells infected with SARS-CoV-2 at the single early time point, 8 hours post infection. While they do not find significant changes to the transcript levels of major metabolites or metabolic pathways, they find significant changes to the nucleotide synthesis pathway. In particular, they show that members of purine synthesis stemming from the pentose phosphate pathway are increased at 8 hours post infection while intracellular glucose levels are decreased. Importantly, this pathway also requires THF which can be added from one carbon metabolism stemming from Folate. They also show that intracellular Folate is decreased in the infected Vero cells, which they propose is being depleted by the need for nucleotide synthesis. While they would require carbon tracing to demonstrate this claim, it is a reasonable hypothesis. They go on to show that Methotrexate, that inhibits DHFR, the first step of metabolizing Folate, leads to decreased virus production and increased infected cell viability. They also show that an inhibitor of a subsequent step, SHIN1 that inhibits SHMT, also decreases virus production and increases cell viability. They then knockout the different SHMTs to show that it is the cytoplasmic enzyme not the mitochondrial enzyme that is required. They propose that MTX, an approved drug in humans, combined with other antivirals could provide a novel treatment for COVID19 patients. Overall, this manuscript is well done and provides important initial results regarding the importance of specific metabolic changes during SARS-CoV-2 infection. While the use of the long-term passaged Vero cells is limiting for some metabolic studies and could limit the identification of some metabolic changes that are induced by SARS-CoV-2 in more relevant primary cells, they have made significant findings in the Vero cells. There are a couple of minor comments to address.

>>>Thanks for the excellent summary and constructive points.

1. In the text they state that “Quantitative production and consumption analyses of spent media metabolites found subtle, mostly non-significant changes between infected and mock-infected cells at this early timepoint.” However, Extended figure 3

appears to show increases in lactate and a couple of other metabolites and slight decreases in glucose, glutamine and pyruvate. Are these not significant? If they are they could be mentioned in the results section.

>>In our spent media analyses, only pyruvate and aspartate show statistically significant changes. Consumption of pyruvate and release of aspartate are significantly increased in SARS-CoV-2-infected cells, albeit mildly with a fold change difference of 0.06 and 0.16 respectively. We now comment on them in the text and we have added p-values to **Extended Figure 2b**.

2. In figure 2A and C and multiple other panels, they use IFA to quantify viral genomic RNA. They do not demonstrate that this is a truly quantitative assay to show viral RNA levels. Could viral RNAs be directly measured using quantitative RT-PCR?

>>We agree that IFA is not truly quantitative, though it does provide a single cell readout and therefore has merit. In the original submission, we provided quantitative RT-PCR-based viral genome copy number and quantitative measurement of viral cytopathic effect. For quantitation of viral subgenomic RNA expression, we developed and now also provide quantitative Np Flow-FISH data for both A549 ACE2 and Vero E6 TMPRSS2 cells. This has the advantage of providing single cell RNA quantification.

Also, they should quantify the percent of infected cells for these experiments since they do not show differences in host transcripts for the metabolic proteins.

>>>The Flow-FISH data added in revision now provides a quantification of infected cells. We also quantitated the % of infected cells in the metabolomic profiling experiments in Figure 1 by analyzing the nucleoprotein signal in multiple IF fields. This data has been added to **Extended Figure 1a**.

3. At the bottom of page 3 they make reference to extended figure 3C but to show the statement made in that sentence they should reference extended figure 3C and 3D

>>Thanks for pointing this out, we fixed this.

Michael Lagunoff

Reviewer #4 (Remarks to the Author):

Summary:

In “SARS-CoV 2 Hijacks Folate and One-Carbon Metabolism for Viral Replication” Zhang et. al. demonstrate the importance of folate and one carbon metabolism for SARS-CoV 2 infection. The overall goal of the paper was to identify key host metabolic pathways modified by SARS-CoV 2 to allow replication. Overall, the paper is well written and the experimental conclusions are consistent with the data. How human CoV manipulate host metabolic pathways is a timely subject matter and the authors’ conclusions will be of interest to a broad audience.

General Points:

As there are several SARS-CoV 2 permissive recombinant human lung epithelial cell lines available why were Vero E6 cells chosen for this study?

>>The use of Vero E6 is advantageous because their high infectivity, which facilitates metabolomic analysis at the early 8 hour timepoint. However, we agree with this point and as written above, we now present new data in **Figures 2j-m, 3d-f and Extended Figs. 5c, 6c and 7b** using the lung epithelial cell line A549, transduced to express the human ACE2 receptor. We used A549 instead of Calu-3 as we could achieve a higher percentage of SARS-CoV-2 infection, and this cell line has been used for transcriptomic profiling by the tenOever lab. We now provide data that SHIN1, methotrexate or swapping the sugar source to galactose each have similar effects in A549 as in Vero E6. Specifically, galactose, SHIN1 and methotrexate protect against SARS-CoV2 CPE, reduce viral load and diminish subgenomic RNA expression.

Specific Points

In Figure 1F it is not always clear which data point the label belongs with. The data point that corresponds with Serine is the most challenging for this reviewer.

>>Thanks for this point. We have now added lines to indicate labeling more clearly.

This reviewer appreciates the amount of time put into capturing the data and making figure 1G, however it is overwhelming. Maybe this could be supplemental and a summary table provided to go with the text. This is only a suggestion.

>>> Thank you for this suggestion. We agree, and the comprehensive diagram was moved to Extended Data. We streamlined new Figure 1G, as shown below:

Figure 2b there is less than 1 log difference in FFU/mL when comparing glucose and galactose. The image and graph show different levels of impairment. Less than one log would not be “strongly impaired”.

>> thanks for this point, we have amended the text to take out the word strongly.

Reviewers' Comments:

Reviewer #1:

Remarks to the Author:

The authors have addressed all of my questions and concerns. There is a striking new result regarding MTX+Brequinar treatment, data is shown on Extended Figure 8. MTX alone has a much bigger effect on suppressing Np subgenomic RNA and viral load compared with MTX+Brequinar. Authors should explain the observed trend; effect of MTX > Brequinar > MTX+ Brequinar. One possible explanation is that a fine balance between purines and pyrimidines intracellular pools is critical for viral RNA replication, MTX alone creates a much bigger nucleotide imbalance compared with MTX+ Brequinar which is costlier for the virus.

Reviewer #2:

Remarks to the Author:

The authors have addressed my main concern of validating their findings in more physiologically relevant cell type.

Reviewer #3:

Remarks to the Author:

In this revision they have added a number of metabolic drug inhibitor studies in the more relevant A459-ACE2 cells. They find similar results in this relevant cell type as they found in the Vero monkey kidney cells indicating the importance of the original results found in these cells. In addition they have added data better demonstrating their infection rates and have altered figures to make them more clear. Therefore, this revision has responded to this reviewers previous comments and has significantly improved the manuscript.

REVIEWERS' COMMENTS

Reviewer #1 (Remarks to the Author):

The authors have addressed all of my questions and concerns. There is a striking new result regarding MTX+Brequinar treatment, data is shown on Extended Figure 8. MTX alone has a much bigger effect on suppressing Np subgenomic RNA and viral load compared with MTX+Brequinar. Authors should explain the observed trend; effect of MTX > Brequinar > MTX+ Brequinar. One possible explanation is that a fine balance between purines and pyrimidines intracellular pools is critical for viral RNA replication, MTX alone creates a much bigger nucleotide imbalance compared with MTX+ Brequinar which is costlier for the virus.

>>thanks for these constructive comments. We speculate that MTX has the strongest effect given its ability to block multiple folate metabolism enzymes and perhaps also reflecting a greater role for purine metabolism in newly SARS-CoV-2 infected cells, as perhaps reflected in the greater *de novo* purine metabolism signal in our metabolomic analysis. We agree that the MTX + Brequinar result is striking. However, we prefer not to speculate on the biological mechanism which is beyond the scope of the current manuscript.

Reviewer #2 (Remarks to the Author):

The authors have addressed my main concern of validating their findings in more physiologically relevant cell type.

>>thanks very much.

Reviewer #3 (Remarks to the Author):

In this revision they have added a number of metabolic drug inhibitor studies in the more relevant A459-ACE2 cells. They find similar results in this relevant cell type as they found in the Vero monkey kidney cells indicating the importance of the original results found in these cells. In addition they have added data better demonstrating their infection rates and have altered figures to make them more clear. Therefore, this revision has responded to this reviewers previous comments and has significantly improved the manuscript.

>>thanks very much.